# Molecular insights into RNA and DNA helicase evolution from the determinants of specificity for a DEAD-box RNA helicase

Anna L Mallam[1,2], David J Sidote[1,2], Alan M Lambowitz[1,2]*

[1]Institute for Cellular and Molecular Biology, University of Texas at Austin, Austin, United States; [2]Department of Molecular Biosciences, University of Texas at Austin, Austin, United States

**Abstract** How different helicase families with a conserved catalytic 'helicase core' evolved to function on varied RNA and DNA substrates by diverse mechanisms remains unclear. In this study, we used Mss116, a yeast DEAD-box protein that utilizes ATP to locally unwind dsRNA, to investigate helicase specificity and mechanism. Our results define the molecular basis for the substrate specificity of a DEAD-box protein. Additionally, they show that Mss116 has ambiguous substrate-binding properties and interacts with all four NTPs and both RNA and DNA. The efficiency of unwinding correlates with the stability of the 'closed-state' helicase core, a complex with nucleotide and nucleic acid that forms as duplexes are unwound. Crystal structures reveal that core stability is modulated by family-specific interactions that favor certain substrates. This suggests how present-day helicases diversified from an ancestral core with broad specificity by retaining core closure as a common catalytic mechanism while optimizing substrate-binding interactions for different cellular functions.

*For correspondence: lambowitz@austin.utexas.edu

Competing interests: The authors declare that no competing interests exist.

## Introduction

Helicases of superfamilies (SFs) 1 and 2 use ATP or other NTPs to bind, unwind, or remodel RNA or DNA in essentially all facets of nucleic acid metabolism (*Preugschat et al., 1996*; *Tanaka and Schwer, 2005*; *Singleton et al., 2007*; *Fairman-Williams et al., 2010*; *Jarmoskaite and Russell, 2014*). They contain a conserved 'helicase core' of two RecA-like domains but act on varied substrates by different mechanisms. SF1 and SF2 helicases can be grouped into families with distinct variations in specificity, mechanism, function, and appended domains (*Figure 1A*) (*Gorbalenya and Koonin, 1993*; *Singleton et al., 2007*; *Fairman-Williams et al., 2010*). The mechanisms by which SF1 and SF2 helicases act on RNA or DNA include non-processive unwinding of short duplexes (e.g., DEAD-box RNA helicases [*Jarmoskaite and Russell, 2011*; *Linder and Jankowsky, 2011*]), unwinding coupled to directional movement ('translocation') along the unwound single strand (e.g., DEAH/RHA, NS3/NPH-II, and RecQ-like helicases [*Pyle, 2008*]), and binding or translocation along a duplex without unwinding (e.g., RIG-I-like and Swi/Snf helicases [*Durr et al., 2005*; *Myong et al., 2009*; *Rawling and Pyle, 2014*]) (*Figure 1A*). How helicases that share a conserved catalytic core evolved such functional diversity remains unknown.

Here, we use the yeast DEAD-box protein Mss116 (*Figure 1B,C*) as a model system to pinpoint the molecular basis for the specificity and mechanism of the conserved helicase core. Mss116 functions as a general RNA chaperone in mitochondrial intron splicing by locally unwinding and disrupting stable but inactive RNA structures that impede RNA folding (*Huang et al., 2005*; *Del Campo et al., 2009*; *Potratz et al., 2011*). As a general RNA chaperone, Mss116 binds diverse RNA substrates non-specifically and has high RNA helicase activity in the absence of partner proteins (*Halls et al., 2007*; *Del Campo et al., 2009*). This makes it an ideal model system to study the properties of an isolated helicase core. The

**eLife digest** Living cells store their genetic material as DNA, which can be copied to make another molecule called RNA. DNA consists of two strands that are wound around each other in a double helix. RNA is made in a similar way to DNA, but it is usually present as a single strand that folds into a three-dimensional structure that is held in shape by regions of the molecule interacting with each other.

Before DNA and RNA can perform their essential tasks in cells, enzymes called helicases must separate the interacting strands. A large group of helicases, known as superfamily 1 and 2, are involved in virtually all aspects of the control of RNA and DNA structure. All of these helicases contain a region called the 'helicase core', but they work in different ways. For example, some move along the DNA or RNA strand whilst they unwind it, while others can unwind RNA without moving. It remains unclear how these helicases have evolved different ways to unwind DNA and RNA structures using the same helicase core.

Mallam et al. have now analyzed a helicase from yeast called Mss116, which belongs to superfamily 2. It is known from previous work that Mss116 binds to many different RNA molecules and—unlike most other helicases—it does not require any extra proteins to help. This makes it an ideal model to study the properties of a helicase core on its own.

Helicases use the energy released from breaking down molecules called nucleotides to pull apart the bonds that hold DNA and RNA strands together. The experiments found that for Mss116, a nucleotide called ATP is the best for providing the energy needed to unwind RNA but other nucleotides can work less efficiently. The experiments also show that in addition to RNA, Ms116 is able to unwind double-stranded DNA molecules that have a certain shape.

Using a technique called X-ray crystallography, Mallam et al. observed the structure of the Mss116 core when it is bound to RNA and DNA. While there are some shared points of contact between the helicase and the DNA or RNA, there are more points of contact between Mss116 and RNA than between Mss116 and DNA.

Mallam et al. propose that present-day helicases have diversified from enzymes that had broad specificity for RNA and DNA, by optimizing interactions that favor the binding of particular nucleotides and nucleic acids. These changes enabled the helicases to become a versatile set of tools that control the structure of RNA or DNA in different ways.

helicase core of Mss116 consists of two RecA-like domains (D1 and D2) that are in an extended 'open state' in the absence of substrates (*Mallam et al., 2011*) and recognize ATP and duplex RNA in a modular manner (*Mallam et al., 2012*) (*Figure 1D*). Upon substrate binding, the two core domains join to form a 'closed state' containing an ATPase active site, while conserved DEAD-box protein motifs in D1 promote the unwinding of short duplexes bound to D2 by excluding one RNA strand and bending the other (*Figure 1D*). The closed-state complex bound to ssRNA and ATP represents the 'post-unwound' state of the helicase core (*Figure 1C*). ATP hydrolysis is required for core reopening and enzyme turnover (*Liu et al., 2008*; *Cao et al., 2011*).

In this study, we determined the structural and biochemical factors that govern how analogues of NTPs (ATP, CTP, GTP, and UTP) and different nucleic acids (single-stranded [ss] RNA, ssDNA, double-stranded [ds] RNA, A-form dsDNA, and B-form dsDNA) interact with the helicase core. In this way, we identify the core–substrate interactions that dictate the physiological specificity and mechanism of Mss116. Our results define the structural and biochemical determinants for the substrate specificity of a DEAD-box protein. Furthermore, they demonstrate that Mss116 has surprisingly ambiguous substrate binding and unwinding properties. Considered in the context of other SF1 and SF2 helicases, our findings show how small structural changes within conserved regions of these protein families can facilitate the emergence of specialized enzymes with new activities and cellular functions.

## Results

### The biochemical basis for the ATP specificity of the helicase core of Mss116

We investigated how Mss116 specifies for ATP during local unwinding by comparing the ability of the helicase core (D1D2, residues 88–597) to use different nucleotides to catalyze RNA unwinding.

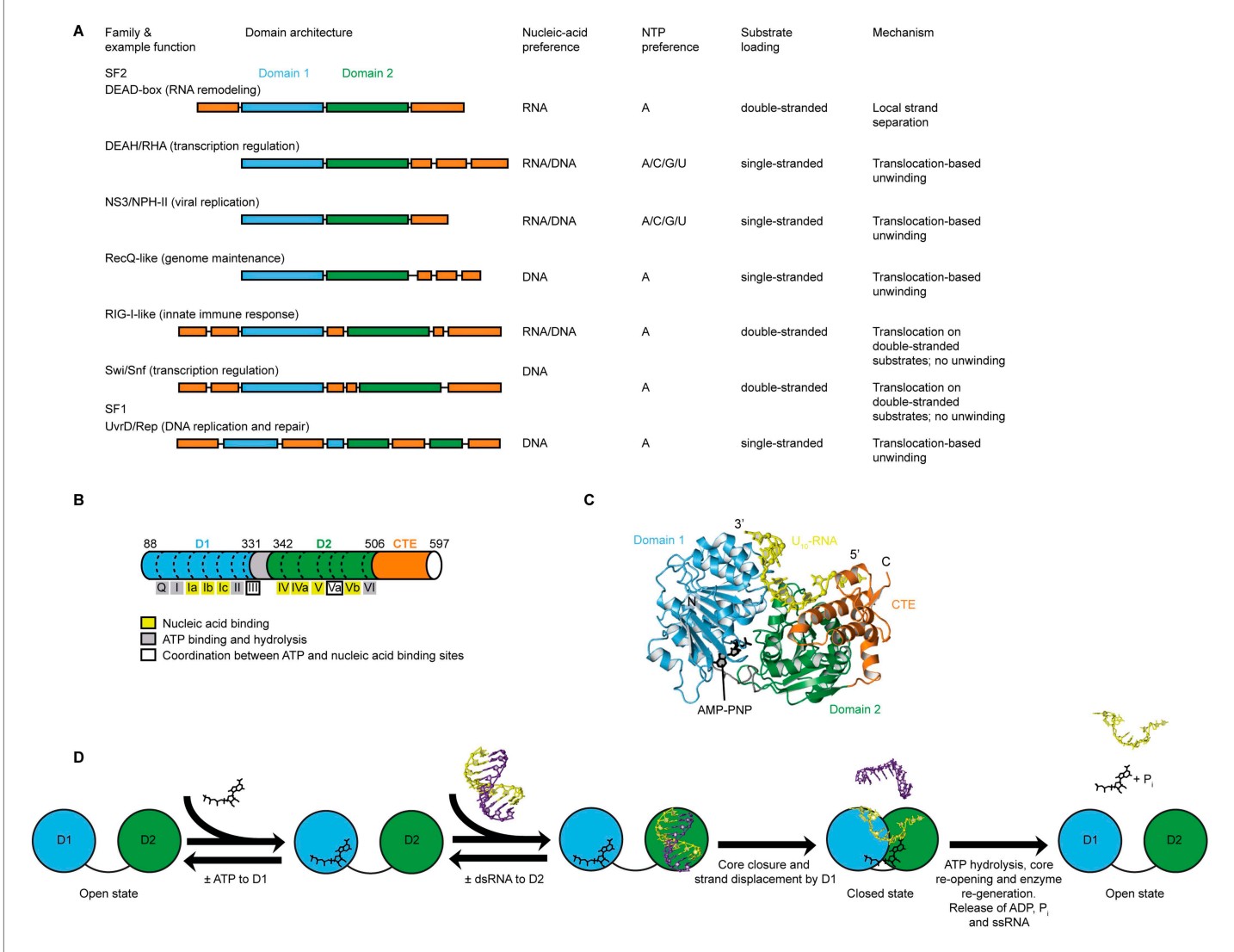

**Figure 1**. Structure, specificity, and mechanisms of the helicase core of Mss116 and other SF1 and SF2 helicases. (**A**) Domain architecture and characteristics of helicases belonging to different SF1 and SF2 families (*Fairman-Williams et al., 2010*). Two other SF1 (Pif1-like and Upf1-like) and four other SF2 (Ski2-like; RecG-like; T1R; and Rad3/XPD) families have been identified (*Fairman-Williams et al., 2010*). Helicase core domains 1 and 2 are colored light blue and green, respectively, while appended domains and insertions, which vary in size, composition, and function, are colored orange; domains are not to scale. (**B**) Schematic of the domain architecture of the helicase core of Mss116 (D1, blue; D2, green; C-terminal extension of D2 [CTE], orange) showing the location of conserved DEAD-box sequence motifs (*Fairman-Williams et al., 2010*). Full-length Mss116 contains additional unstructured N-terminal (residues 37–87) and C-terminal (residues 598–664) extensions that are not required for helicase activity (*Cao et al., 2011*; *Mohr et al., 2011*). (**C**) Structure of the closed-state helicase core of Mss116 (PDB accession 3I5X) (*Del Campo and Lambowitz, 2009*) bound to ssRNA (U10-RNA; yellow) and adenosine nucleotide (AMP-PNP; black). (**D**) Model for RNA duplex binding and unwinding by Mss116. The helicase core domains of Mss116 have modular roles in substrate loading (*Mallam et al., 2012*). D1 captures ATP in the open-state enzyme using the Q-motif, which coordinates the adenine base, and motifs I and II, which are the conserved triphosphate-binding loop and $Mg^{2+}$-binding aspartic acid motifs, respectively, present in many other ATP-binding enzymes (*Walker et al., 1982*; *Rudolph et al., 2006*; *Schutz et al., 2010*; *Mallam et al., 2012*). D2 recognizes duplex RNA (*Mallam et al., 2012*). When ATP and dsRNA are bound to D1 and D2, respectively, core closure occurs, leading to unwinding of the dsRNA bound to D2 by bending one RNA strand and displacing the other. During unwinding and formation of the closed-state helicase core complex, ATP bound to D1 makes additional interactions with motifs Va and VI in D2. The closed-state helicase core bound to ssRNA and ATP represents the 'post-unwound' state of the enzyme (*Figure 1C*). ATP hydrolysis occurs in the closed state, followed by dissociation of $P_i$ and ADP, which leads to the reopening of the core and the release of the bound ssRNA, thereby regenerating the enzyme (*Henn et al., 2010*; *Cao et al., 2011*).

The following figure supplement is available for figure 1:

**Figure supplement 1**. Crystal structures of helicases belonging to different SF1 and SF2 families.

First, we measured the concentration of different NTP analogues required by the helicase core to unwind an RNA duplex under equilibrium conditions (*Figure 2A*). This was done by using a 12-base pair (bp) dsRNA, which was labeled with a fluorophore and quencher at its 5' and 3' ends, respectively. A native gel-based assay was then used to monitor unwinding by the increase in fluorescence in a closed-state core containing a bound single strand (*Figure 2—figure supplement 1*). We find that all of the non-hydrolyzable analogues NDP-BeF$_x$, where N = A, C, G, or U, can promote the unwinding of a dsRNA. However, ADP-BeF$_x$ is the most efficient with at least sixfold higher concentrations of C-, G-, or U-analogues required for RNA duplex unwinding ($K_{1/2}$ = 0.14, 0.8, 0.8, and 2.4 mM, respectively; *Figure 2A* and *Figure 2—figure supplement 1B–E*).

Kinetic unwinding assays were also performed using the same dye-labeled dsRNA in the presence of an unlabeled duplex. In these experiments, an increase in fluorescence occurs upon unwinding of a labeled duplex and subsequent re-annealing to an unlabeled strand. This was measured by isolating the duplexes using native gel electrophoresis at various times after unwinding was initiated by the addition of NTP, where N = A, C, G, or U (*Figure 2—figure supplement 2*). These assays show that only ATP, and not other NTPs, catalyzes the unwinding of the dsRNA (*Figure 2—figure supplement 2B–D*). This indicates that under our assay conditions, the diphosphate beryllium fluoride analogue is necessary to promote unwinding with nucleotide bases other than adenine. This difference likely reflects that the

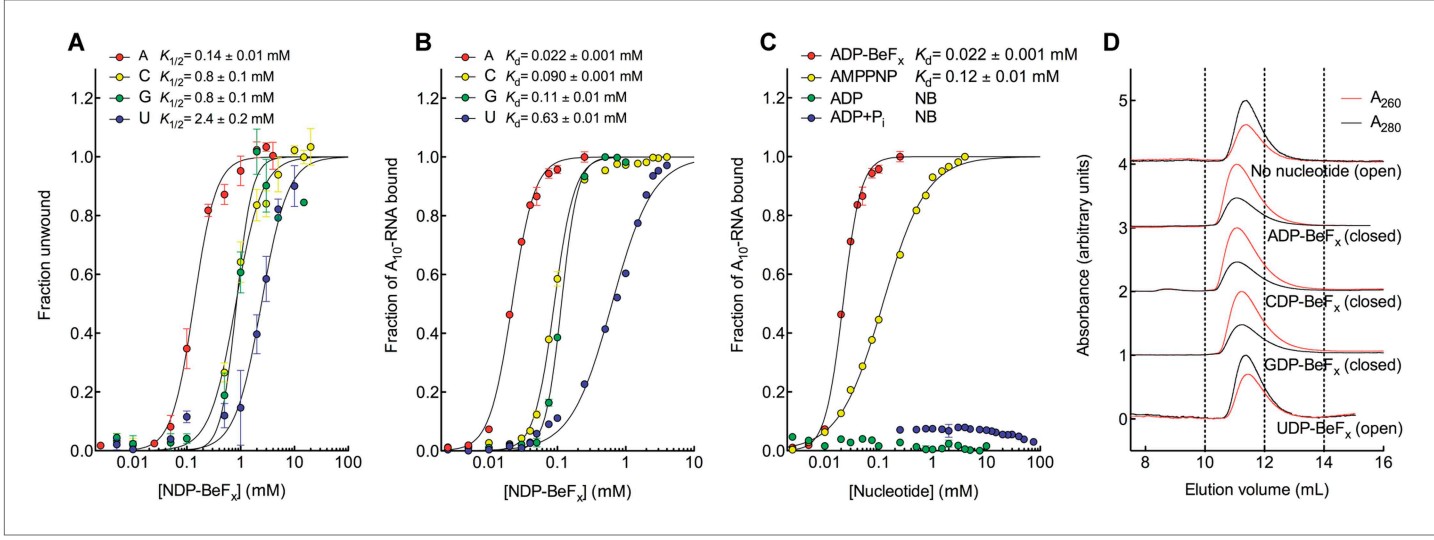

**Figure 2**. The biochemical basis for the ATP specificity of the helicase core of Mss116. (**A**) dsRNA unwinding by the MBP-tagged helicase core measured under equilibrium conditions using a gel-based fluorescence assay to monitor the formation of a closed-state complex containing bound ssRNA at increasing concentrations of NDP-BeF$_x$, N = A, C, G, or U (*Figure 2—figure supplement 1*). The fraction of unwound duplex was obtained by normalizing the band intensities separately for each gel using the parameters from the fit to a one-site binding model, as the change in fluorescence upon unwinding is different under each condition. The extent of unwinding with UDP-BeF$_x$ was less than that for the other nucleotide analogs, and the maximum concentration of UDP-BeF$_x$ used in this assay was insufficient to drive unwinding to completion (*Figure 2—figure supplement 1*). This could be because UDP-BeF$_x$ bound at saturating concentrations to D1 cannot efficiently induce a closed state. (**B**) Equilibrium binding of A$_{10}$-RNA to the MBP-tagged helicase core determined by fluorescence anisotropy measurements at increasing concentrations of NDP-BeF$_x$, N = A, C, G, or U. (**C**) Equilibrium binding of A$_{10}$-RNA to the MBP-tagged helicase core determined as in (**B**) at increasing concentrations of ADP-BeF$_x$, AMP-PNP, ADP, and ADP + P$_i$. Error bars in (**A–C**) represent the standard error for at least three independent measurements, and the error in the $K_{1/2}$ or $K_d$ represents the standard error of the non-linear regression. NB, no appreciable binding. In (**B** and **C**), the fraction of A$_{10}$-RNA bound was calculated by normalizing against the anisotropy signal for unbound and fully bound substrate obtained from the fit to a one-site binding model. (**D**) Normalized SEC profiles monitored by absorbance at 260 nm (red) and 280 nm (black) for the helicase core in the absence of all substrates and in the presence of A$_{10}$-RNA + NDP-BeF$_x$, N = A, C, G, or U. An A$_{260}$/A$_{280}$ >1 at the maximum absorbance indicates the formation of a closed-state complex.

The following figure supplements are available for figure 2:

**Figure supplement 1**. RNA unwinding measured by using a gel-based fluorescence assay to monitor the formation of a closed-state complex containing bound ssRNA.

**Figure supplement 2**. Kinetic assay of the unwinding of dsRNA by Mss116 with different NTPs.

NDP-BeF$_x$ analogues form longer-lived, more stable complexes with RNA than do the corresponding NTPs (*Liu et al., 2014*).

We next examined how the stability of the ternary closed-state complex with ssRNA and the same NTP analogues correlates with the efficiency of duplex unwinding. Equilibrium fluorescence anisotropy binding assays with a fluorescein (FAM)-labeled A$_{10}$-RNA were used to monitor formation of the closed state with increasing concentrations of NDP-BeF$_x$ (N = A, C, G, or U; *Figure 2B*). These assays show that the closed-state complex is most stable with ADP-BeF$_x$ ($K_d$ = 0.022 mM), while CDP-BeF$_x$, GDP-BeF$_x$, and UDP-BeF$_x$ promote formation of the closed state only at significantly higher concentrations of nucleotide analogue ($K_d$ = 0.09, 0.11, and 0.63 mM, respectively). Similarly, analytical size-exclusion chromatography (SEC) shows that a closed-state helicase core with A$_{10}$-RNA is maintained during elution for complexes containing ADP-BeF$_x$, CDP-BeF$_x$, or GDP-BeF$_x$ but not those containing UDP-BeF$_x$, consistent with the latter complex having a lower stability (*Figure 2D* and *Table 1*). Together, these findings indicate that the unwinding efficiencies and closed-state core stabilities with different NTP analogues follow the same order of A > C, G > U from higher to lower efficiency and stability.

Additional fluorescence anisotropy assays show that a closed-state complex with A$_{10}$-RNA forms at significantly lower concentrations of ADP-BeF$_x$ compared to AMP-PNP ($K_d$ = 0.022 and 0.12 mM, respectively; *Figure 2C*). This indicates a more stable closed state and accounts for the higher unwinding activity observed for ADP-BeF$_x$ compared to AMP-PNP for several DEAD-box proteins (*Liu et al., 2008*). Further, neither ADP nor ADP + P$_i$ in large excess led to the formation of a stable closed state in our assays (*Figure 2C*), suggesting that the effective concentration of the ATP γ-phosphate is critical for the stability of the closed-state. This finding explains energetically why ATP hydrolysis leads to core re-opening and enzyme turnover in DEAD-box proteins (*Henn et al., 2010*; *Cao et al., 2011*) and perhaps other SF1 and SF2 helicases. Together, our results show the unwinding efficiency of Mss116 with different nucleotides is directly correlated with the stability of the post-unwound closed-state complex.

## The structural basis for the ATP specificity of the helicase core of Mss116

To investigate the structural basis for the difference in stability of the closed state with different NTP analogs, we determined crystal structures of the closed-state helicase core with A$_{10}$-RNA and either

**Table 1.** Size exclusion chromatography analysis of the helicase core of Mss116

| Sample | Elution volume at maximum absorbance/ml | A$_{260}$/A$_{280}$ of peak at maximum absorbance | Likely predominant state of core |
|---|---|---|---|
| Free protein | | | |
| D1D2 (Mss116 helicase core) | 11.4 | 0.6 | Open |
| Free nucleic acid | | | |
| dsRNA | 16.3 | 2.1 | – |
| A-DNA duplex | 15.1 | 1.6 | – |
| B-DNA duplex | 15.3 | 1.9 | – |
| A$_{10}$-RNA | 18.6 | 3.0 | – |
| A$_{10}$-DNA | 16.6 | 3.4 | – |
| Protein–RNA–nucleotide complexes* | | | |
| D1D2–dsRNA–ADP-BeF$_x$ | 9.6 | 1.1 | Closed |
| D1D2–A-DNA-duplex–ADP-BeF$_x$ | 9.6 | 1.2 | Closed |
| D1D2–B-DNA-duplex–ADP-BeF$_x$ | 11.4 | 0.6 | Open |
| D1D2–A$_{10}$-RNA–ADP-BeF$_x$ | 11.0 | 2.2 | Closed |
| D1D2–A$_{10}$-RNA–CDP-BeF$_x$ | 11.1 | 2.2 | Closed |
| D1D2–A$_{10}$-RNA–GDP-BeF$_x$ | 11.2 | 2.3 | Closed |
| D1D2–A$_{10}$-RNA–UDP-BeF$_x$ | 11.4 | 1.0 | Open |
| D1D2–A$_{10}$-DNA–ADP-BeF$_x$ | 11.4 | 0.6 | Open |

*Parameters are quoted for the peak containing protein as determined by A$_{214}$.

ADP-BeF$_x$, CDP-BeF$_x$, GDP-BeF$_x$, or UDP-BeF$_x$ at 2.2, 2.7, 2.4, and 3.2 Å resolution, respectively (*Figure 3* and *Table 2*). These structures show that the ATP-binding motifs I and VI make similar direct contacts to the phosphate groups of all four NTP analogs (*Figure 3C*). Motif II (DEAD) is positioned identically in all structures and interacts indirectly via waters with the BeF$_3$ moiety, which corresponds to the ATP γ-phosphate (*Figure 3B*). However, each base interacts differently in the ATP-binding pocket. The purine bases (A and G) are stacked optimally with F126 in the Q-motif, which primarily confers ATP specificity in DEAD-box proteins (*Linder and Jankowsky, 2011*), whereas the pyrimidine bases (C and U) adopt a less favorable stacking orientation with this residue (*Figure 3B*). Also, fewer direct contacts are made to the C, G, and U bases than to A (*Figure 3C*). In particular, compared to the closed-state structure with ADP-BeF$_x$, two hydrogen (H)-bonds from G128 and Q133 in the Q-motif to the base are absent in the complex with GDP-BeF$_x$, and all of the direct interactions of the Q-motif with the base are missing in the structures with CDP- or UDP-BeF$_x$. The fewer contacts of all other bases relative to adenine and the less favorable stacking of pyrimidine bases in the ATP-binding pocket explain the relative stabilities of the closed-state complexes and reveal how the helicase core of Mss116 adapted to unwind RNA most efficiently using ATP.

## The biochemical basis for the RNA specificity of the helicase core of Mss116

D2 of Mss116 (residues 342–597) functions as an RNA-duplex recognition domain in the open-state enzyme (*Mallam et al., 2012*) (*Figure 1D*). To determine how Mss116 specifies for dsRNA, we first

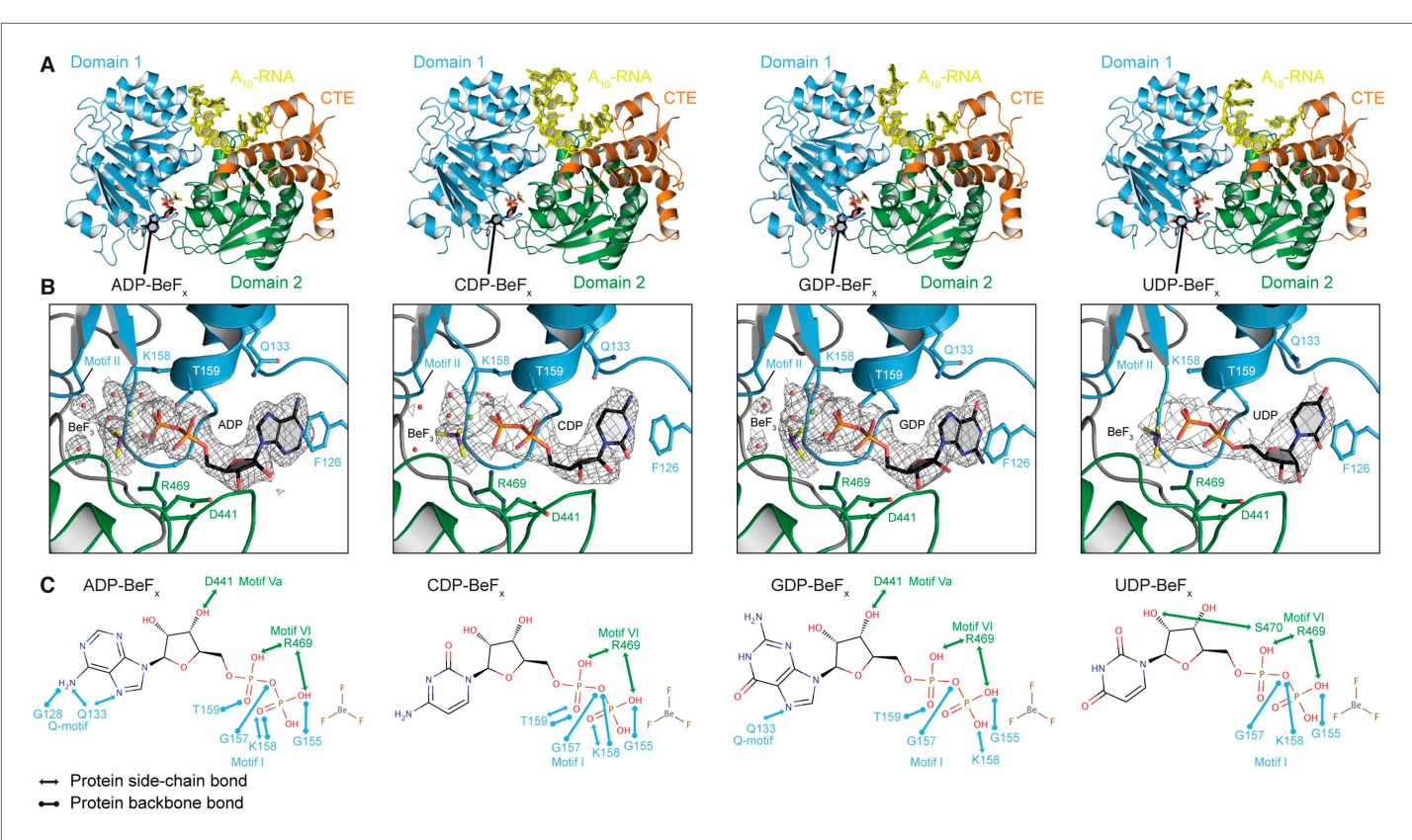

**Figure 3**. The structural basis for the ATP specificity of the helicase core of Mss116. (**A**) Crystal structures of the closed-state helicase core of Mss116 bound to ssRNA and different nucleotide analogues (D1D2–A$_{10}$-RNA–NDP-BeF$_x$ for N = A, C, G, or U). Structures are colored according to the scheme in *Figure 1C*. (**B**) Comparisons of the protein–substrate interactions in the NDP-BeF$_x$ binding pockets of the structures shown in (**A**). Side chains that make direct contacts with the NDP are shown as ball and stick models. A 2F$_o$ − F$_c$ electron density map contoured at 1.0 σ for the NDP-BeF$_x$ ligand is shown in gray. Mg$^{2+}$ ions and water molecules are shown as green and red spheres, respectively, and the atoms of BeF$_3$ are shown as purple (Be) and yellow (F). Motif II ('DEAD') makes indirect contacts via water molecules to the BeF$_3$ moiety, which corresponds to the γ-phosphate of ATP. (**C**) Schematics of direct NDP–protein interactions for the structures shown in (**A**). See also *Table 2*.

**Table 2.** Crystallographic data and refinement statistics

| Complex | D1D2–A$_{10}$-RNA–ADP-BeF$_x$ | D1D2–A$_{10}$-RNA–CDP-BeF$_x$ | D1D2–A$_{10}$-RNA–GDP-BeF$_x$ | D1D2–A$_{10}$-RNA–UDP-BeF$_x$ | D1D2–A$_{10}$-DNA–ADP-BeF$_x$ |
|---|---|---|---|---|---|
| Data collection | | | | | |
| Space group | P2$_1$2$_1$2 | P2$_1$2$_1$2 | P2$_1$2$_1$2 | P2$_1$2$_1$2 | P2$_1$2$_1$2 |
| Unit cell | | | | | |
| a, b, c (Å) | 89.83, 126.26, 55.55 | 89.64, 126.84, 55.03 | 89.99, 126.61, 55.55 | 89.76, 126.51, 55.51 | 90.39, 126.19, 55.23 |
| α, β, γ (°) | 90, 90, 90 | 90, 90, 90 | 90, 90, 90 | 90, 90, 90 | 90, 90, 90 |
| Wavelength (Å) | 1.0000 | 1.0000 | 1.0000 | 1.0000 | 1.0000 |
| Total reflections | 222,375 | 129,478 | 565,729 | 86,580 | |
| Unique reflections | 32,642 | 16,982 | 27,148 | 10,514 | 13,111 |
| Resolution* (Å) | 50 − 2.20 (2.24 − 2.20) | 50 − 2.60 (2.64 − 2.60) | 50 − 2.35 (2.39 − 2.35) | 50 − 3.30 (3.36 − 3.30) | 50 − 3.00 (3.05 − 3.00) |
| Redundancy | 6.8 (5.4) | 6.1 (5.4) | 19.2 (14.1) | 8.2 (8.1) | 5.5 (4.9) |
| Completeness (%) | 99.4 (97.7) | 99.7 (98.3) | 99.5 (94.9) | 99.5 (95.0) | 96.7 (88.8) |
| Overall $I/\sigma(I)$ | 19.0 (1.5) | 12.1 (1.5) | 26.4 (2.4) | 11.1 (2.5) | 7.1 (1.5) |
| R$_{merge}$† (%) | 9.7 (60.3) | 13.8 (77.0) | 13.3 (99.7) | 19.8 (66.5) | 19.8 (61.2) |
| Refinement | | | | | |
| Resolution (Å) | 47.24 − 2.20 | 47.07 − 2.74 | 47.28 − 2.35 | 47.22 − 3.21 | 44.15 − 2.96 |
| No. of reflections | 32,642 | 16,982 | 27,146 | 10,514 | 13,111 |
| R$_{work}$ (%) | 21.6 | 22.3 | 23.09 | 22.16 | 19.6 |
| R$_{free}$§ (%) | 25.4 | 26.7 | 26.00 | 27.43 | 24.4 |
| No. atoms | | | | | |
| Protein | 7911 | 7519 | 7723 | 7528 | 7774 |
| Nucleic acid | 232 | 298 | 230 | 232 | 147 |
| Ligands | 45 | 42 | 43 | 40 | 44 |
| Water | 115 | 28 | 61 | 0 | 0 |
| Rmsd bonds (Å) | 0.003 | 0.003 | 0.003 | 0.004 | 0.003 |
| Rmsd angles (°) | 0.696 | 0.629 | 0.710 | 0.968 | 0.751 |
| Ramachandran favored# (%) | 97.23 | 96.01 | 96.84 | 98.40 | 97.04 |
| Ramachandran allowed (%) | 2.30 | 1.94 | 2.19 | 1.40 | 1.98 |
| PDB ID | 4TYW | 4TYY | 4TZ0 | 4TZ6 | 4TYN |

*The numbers in parentheses refer to the highest resolution shell.

†R$_{merge}$ = $\sum_{hkl} \sum_i |I_{hkl,i} − \langle I_{hkl} \rangle| / \sum_{hkl} \sum \langle I_{hkl} \rangle$.

§R$_{free}$ was calculated with 5% of reflections that were excluded from refinement.

#Analysis by MolProbity (**Chen et al., 2010**).

examined how 12-bp RNA and DNA duplexes of different geometries (**Figure 4**) interact with D2 in the absence of nucleotide. EMSAs using fluorescein amidite (FAM)-labeled duplexes show that D2 has surprisingly similar affinities for dsRNA ($K_{1/2}$ = 400 nM) and A-DNA and B-DNA duplexes ($K_{1/2}$ = 410 and 510 nM, respectively) (**Figure 5A** and **Figure 5—figure supplement 1A–C**). Circular dichroism (CD) measurements confirmed that the geometry of the A-DNA and B-DNA duplexes is maintained upon binding to D2, and that binding does not induce a B- to A-form transition (**Figure 4E** and **Figure 4—figure supplement 1**). The B-DNA duplex also competitively displaces dsRNA bound to D2

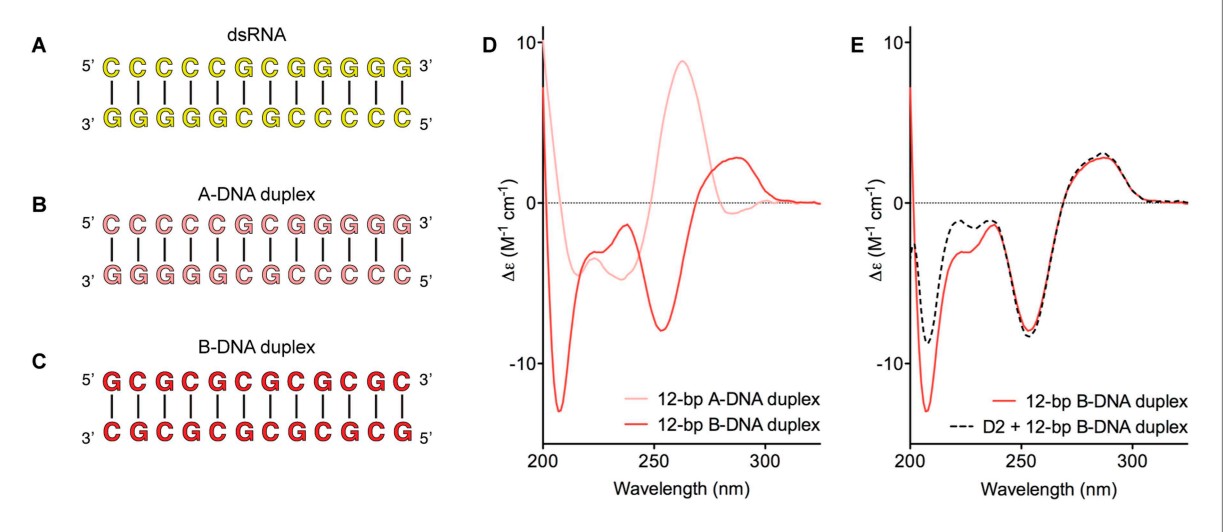

**Figure 4**. Model nucleic acid substrates. (**A**–**C**) 12-bp model substrates of (**A**) dsRNA (yellow); (**B**) A-DNA duplex (pink); and (**C**) B-DNA duplex (red). The duplex geometry of the DNA substrates has been previously characterized in solution by CD measurements (**Basham et al., 1995**) and X-ray crystallography (**Verdaguer et al., 1991**). The duplexes are predicted to have similar stabilities (predicted melting temperatures are 61.0°C, 59.4°C, and 63.9°C for the dsRNA, A-DNA, and B-DNA duplexes, respectively [**Owczarzy et al., 2008**]). (**D**) CD spectra of A-DNA (pink) and B-DNA (red) duplexes, which are consistent with previously reported spectra of identical duplexes (**Basham et al., 1995**; **Kypr et al., 2009**). The CD-spectrum of the A-DNA duplex has a characteristic strong positive peak at 260 nm and negative peaks at 240 and 210 nm (**Ivanov et al., 1973**). The B-DNA duplex is characterized by a positive peak at 260–280 nm and a negative peak at ~245 nm (**Kypr et al., 2009**). (**E**) CD spectra of the B-DNA duplex (100 μM) in the absence (solid red line) and presence (dashed black line) of D2 (120 μM). Spectra are shown in units of molar circular dichroism (Δε) and are background subtracted for the presence of protein.

The following figure supplement is available for figure 4:

**Figure supplement 1**. CD spectra of A-DNA duplex (80 μM) in the absence (solid pink line) and presence (dashed black line) of MBP-D2 (100 μM).

($K_i$ = 1700 nM) (**Figure 5B**). These results indicate that D2 can bind dsRNA and dsDNA of A- or B-form geometry in the dsRNA binding pocket even with the different spacing of the backbone phosphate groups (**Mallam et al., 2012**). Our findings are consistent with recent studies showing that several DEAD-box proteins can interact with dsDNA (**Kammel et al., 2013**; **Tuteja et al., 2014**). D2 of Mss116 is therefore a general and flexible nucleic acid duplex binding domain.

We next examined the ability of Mss116 to unwind the same RNA and DNA model duplexes in the presence of increasing concentrations of ADP-BeF$_x$ (**Figure 5C** and **Figure 5—figure supplement 2**). Equilibrium duplex unwinding assays (**Figure 2—figure supplement 1A**) show that Mss116 can unwind dsRNA and an A-DNA duplex, although a lower concentration of ADP-BeF$_x$ is required to unwind dsRNA ($K_{1/2}$ = 0.14 and 0.25 mM, respectively). Notably, we did not observe any appreciable unwinding of the B-DNA duplex under these conditions (**Figure 5C** and **Figure 5—figure supplement 2**). In this case, kinetic unwinding assays demonstrate the same trend. They show that Mss116 can unwind dsRNA and the A-DNA duplex in the presence of ATP with observed first-order rate constants ($k_1$) of 0.46 and 0.15 min$^{-1}$, respectively, but does not unwind the B-DNA duplex (**Figure 5—figure supplement 3**). Similarly, analytical SEC showed elution profiles for D1D2 that are consistent with closed-state complexes when measured with ADP-BeF$_x$ and dsRNA or the A-DNA duplex but not the B-DNA duplex (**Table 1** and **Figure 5—figure supplement 4**). These data indicate that Mss116 selectively unwinds A-form duplex nucleic acids. Further, contrary to what was previously thought (**Fairman-Williams et al., 2010**), they demonstrate that a DEAD-box protein can unwind an all DNA duplex in a nucleotide-dependent manner if it has A-form geometry. Although D2 can bind a B-DNA duplex, a closed-state complex does not readily form with B-form DNA and unwinding of this substrate does not occur.

To further investigate why Mss116 preferentially unwinds RNA duplexes, we compared the characteristics of the closed-state helicase core with equivalent ssRNA (A$_{10}$-RNA) and ssDNA (A$_{10}$-DNA) substrates. Equilibrium fluorescence anisotropy assays in the presence of increasing concentrations of ADP-BeF$_x$

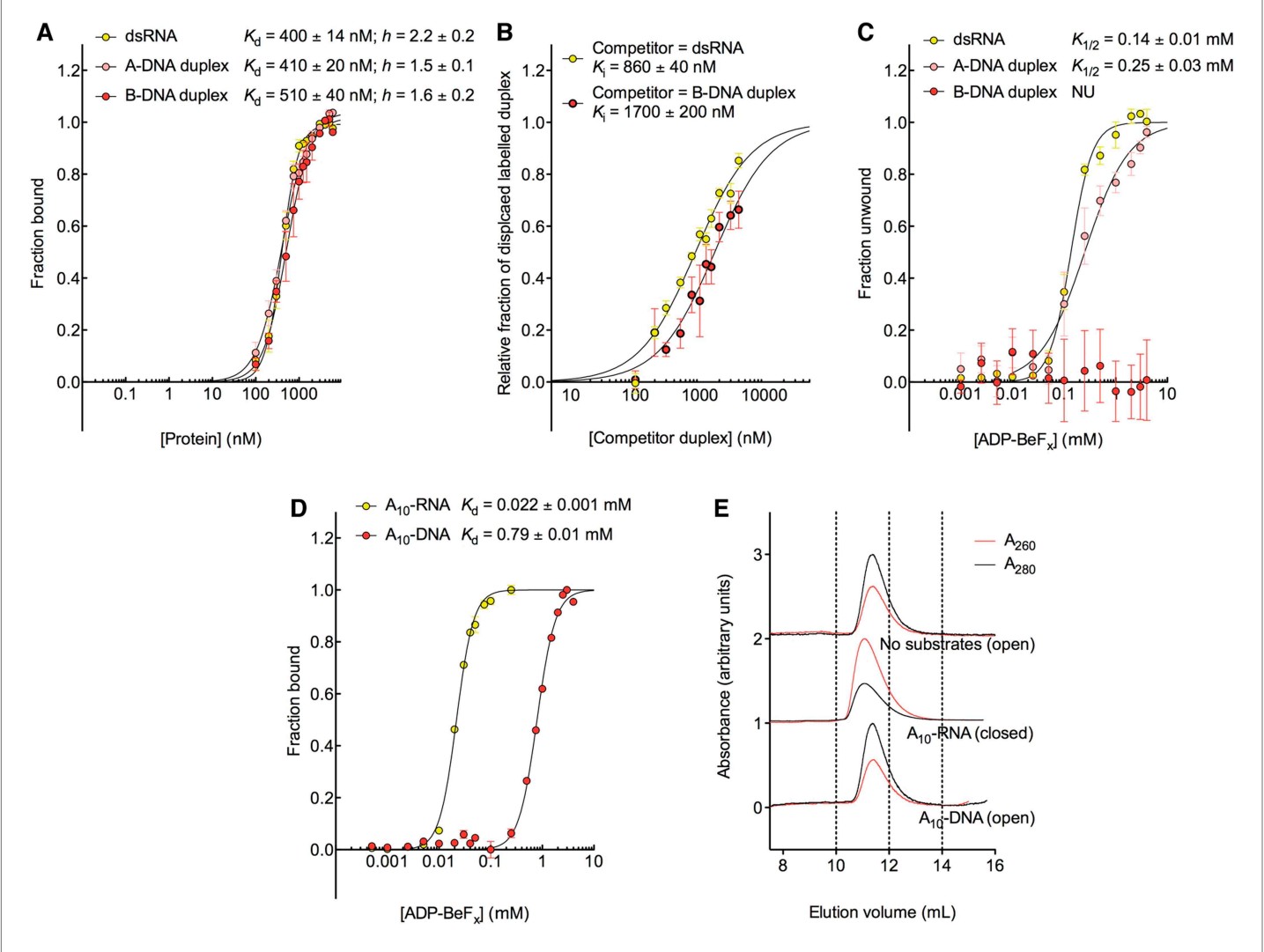

**Figure 5**. The biochemical basis for the RNA specificity of the helicase core of Mss116. (**A**) Equilibrium binding of duplex substrates to MBP-tagged D2 in the absence of nucleotide determined by EMSA. (**B**) Competitive displacement from MBP-tagged D2 of 5' FAM-B-DNA duplex (250 nM) by unlabeled dsRNA (0–6 μM, yellow, $K_i$ = 860 ± 40 nM) and of 5' FAM-dsRNA (250 nM) by unlabeled B-DNA duplex (0–6 μM, red, $K_i$ = 1700 ± 200 nM). (**C**) Unwinding of duplex substrates by the MBP-tagged helicase core measured under equilibrium conditions by using a gel-based fluorescence assay to monitor the formation of a closed-state complex at increasing concentrations of ADP-BeF$_x$ (see also **Figure 2—figure supplement 1**). NU, no appreciable unwinding. (**D**) Equilibrium binding of A$_{10}$-DNA to the MBP-tagged helicase core determined by fluorescence anisotropy measurements at increasing concentrations of ADP-BeF$_x$. The binding of A$_{10}$-RNA under the same conditions is shown for comparison (taken from **Figure 2B**). In (**A–D**), data were normalized using the signal obtained from the fit to the appropriate model outlined in the 'Materials and methods'. (**E**) Normalized SEC profiles monitored by the absorbance at 260 nm (red) and 280 nm (black) for the helicase core in the absence of substrates (top) and in the presence of either A$_{10}$-RNA + ADP-BeF$_x$ (middle) and A$_{10}$-DNA + ADP-BeF$_x$ (bottom). An A$_{260}$/A$_{280}$ >1 at the maximum absorbance indicates the formation of a stable closed-state complex (**Table 1**).

The following figure supplements are available for figure 5:

**Figure supplement 1**. EMSA binding assays of model duplexes.

**Figure supplement 2**. Duplex unwinding measured by using a gel-based fluorescence assay to monitor the formation of a closed-state complex containing bound ssRNA or ssDNA.

**Figure supplement 3**. Kinetic assay of unwinding of duplex substrates by ATP.

**Figure supplement 4**. Characterization of the helicase core in the absence and presence of duplex substrates using size-exclusion chromatography.

indicate that the closed-state complex forms with both substrates, but at a much lower concentration of ADP-BeF$_x$ for ssRNA than for ssDNA ($K_d$ = 0.022 and 0.79 mM, respectively; *Figure 5D*). SEC data also demonstrate that a closed-state complex with A$_{10}$-RNA and ADP-BeF$_x$ remains intact during elution, whereas an identical complex with A$_{10}$-DNA dissociates on the SEC column (*Figure 5E* and *Table 1*). Thus, the closed-state core is significantly more stable and long-lived with ssRNA than with ssDNA.

### The structural basis for the RNA specificity of the helicase core of Mss116

To probe the structural basis for the difference in stability of the closed-state complex with ssRNA compared to ssDNA, we determined crystal structures of the closed-state helicase core with ADP-BeF$_x$ and either A$_{10}$-RNA or A$_{10}$-DNA at 2.5 and 2.9 Å resolutions, respectively (*Figure 6* and *Table 2*). These structures confirm that Mss116 can form the same closed-state complex with ssRNA and ssDNA and allow a direct comparison of the interactions made by these substrates with the same helicase core. The structures show that trajectories of the bound ssRNA and ssDNA are very similar (*Figure 6B*) and that most of the interactions between the conserved nucleic acid binding motifs IV–V and the phosphate backbone are identical in both complexes (*Figure 6C*). However, the closed-state complex with ssRNA contains protein contacts to RNA 2′-OH groups that are not present in the closed-state complex with ssDNA. These include four from residues in motifs Ia and Ic in D1 that form during core closure and account for the higher stability of the closed-state with ssRNA (*Figure 5D,E*).

### Discussion

Collectively our results elucidate the basis for the physiological preference of the DEAD-box protein Mss116 for ATP and RNA, but also show that the helicase core has a surprising degree of substrate ambiguity. This is a consequence of the ability of conserved helicase motifs to interact with the phosphate groups of different NTPs or nucleic acids and promote the formation of the same closed-state

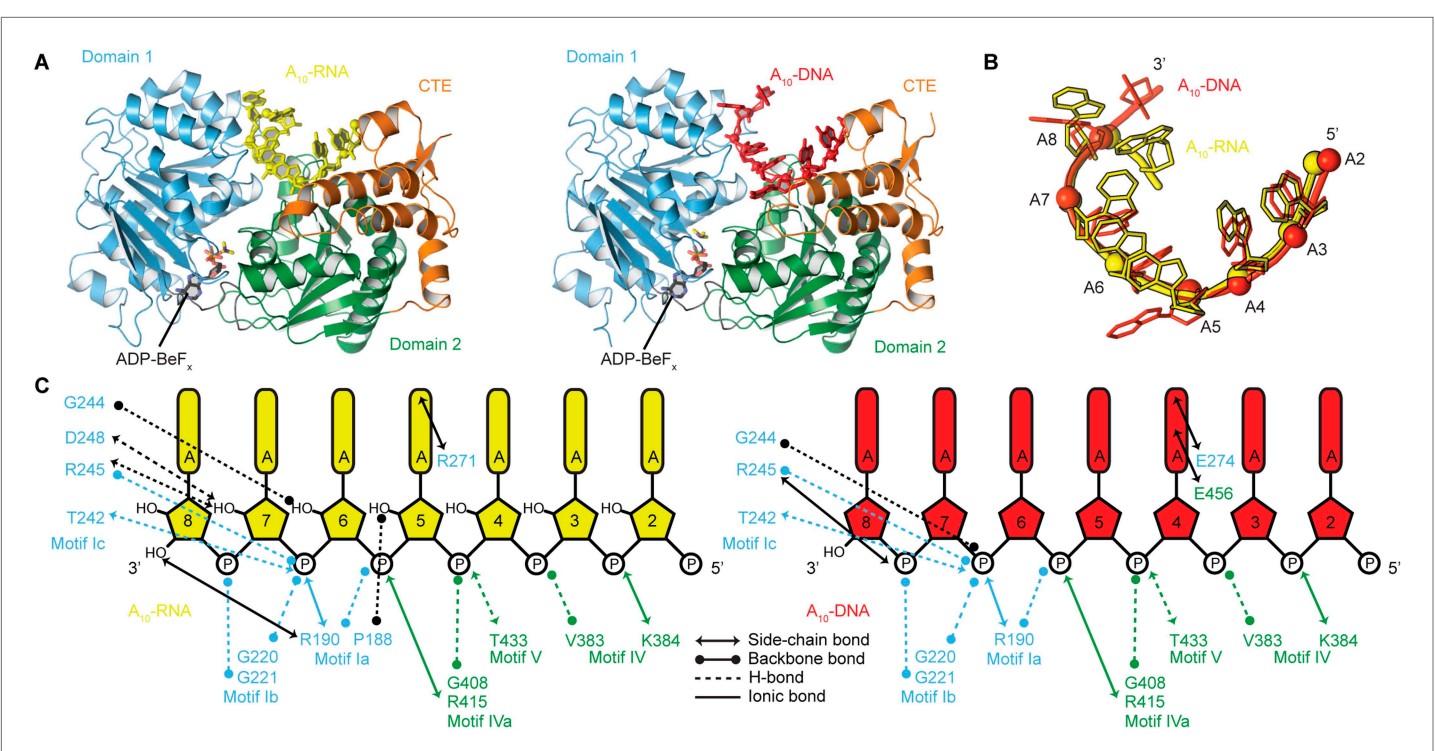

**Figure 6**. The structural basis for the RNA specificity of the helicase core of Mss116. (**A**) Closed state crystal structures of the helicase core of Mss116 with the ATP analogue ADP-BeF$_x$ and A$_{10}$-RNA (yellow) or A$_{10}$-DNA (red). The helicase core is colored as in *Figure 1C*. (**B**) A comparison of the binding trajectory of equivalent nucleotides of A$_{10}$-RNA (yellow) and A$_{10}$-DNA (red) bound in the closed state. (**C**) A schematic comparing the interactions of A$_{10}$-RNA (yellow) and A$_{10}$-DNA (red) with the closed-state helicase core, colored blue and green to D1 and D2, respectively. Interactions unique to each structure are colored black.

complex (*Figure 3A* and *Figure 6A*). The preference of Mss116 for ATP is dictated by optimal base-stacking and H-bonding interactions between the Q-motif and adenine base (*Figure 3B,C*). However, interactions between conserved motifs I, II, and VI and nucleotide phosphate moieties are sufficient to promote duplex unwinding at lower efficiency irrespective of the nucleotide base (*Figure 2A* and *Figure 3B,C*).

The specificity of Mss116 for unwinding RNA duplexes is dictated by both A-form geometry (*Figure 5C*) and interactions by motifs Ia and Ic in D1 with 2′-OH groups of ssRNA in the closed state (*Figure 5D* and *Figure 6C*). Additionally, Mss116 belongs to a subclass of DEAD-box proteins that has a CTE appended to D2 (*Figure 1B*) (*Mohr et al., 2008*). This CTE makes additional 2′-OH contacts to dsRNA in the open state (*Mallam et al., 2012*) that may favor its binding to D2 (*Figure 5B*). Nevertheless, the interactions of nucleic acid-binding motifs with the phosphate backbone are sufficient to enable Mss116 to unwind A-form DNA duplexes at lower efficiency (*Figure 5C* and *Figure 5—figure supplement 3*). Mss116 cannot unwind a B-form DNA duplex (*Figure 5C* and *Figure 5—figure supplement 3*), and a model of the closed state with a B-DNA duplex indicates that the helicase motifs in D1 that clash with dsRNA (*Mallam et al., 2012*) (*Figure 7A*) are not positioned to catalyze the unwinding of longer, thinner B-form duplexes (*Figure 7B*).

Importantly, the substrate ambiguity of Mss116 suggests an evolutionary scenario for how SF1 and SF2 helicases diverged from an ancestral helicase core with broad specificity into specialized enzymes. In each case, core closure was retained as a catalytic mechanism using the interactions common to all NTP or nucleic acid substrates predicted from our results. However, the stability of the closed-state was further modulated by family-specific interactions that favor a particular NTP and nucleic acid. Thus, helicase families that display the most substrate ambiguity by utilizing all four NTPs and function on either DNA or RNA (for example the DEAH/RHA [*Tanaka and Schwer, 2005*] and NS3/NPH-II [*Preugschat et al., 1996*] families; *Figure 1A*) may contain a core that functions similarly to that of an ancestral helicase. Helicases that preferentially use ATP maintained the conserved interactions with nucleotide phosphate groups but acquired additional interactions with the adenine base that further stabilize the closed-state complex. Similarly, DEAD-box proteins, which act preferentially on RNA (*Fairman-Williams et al., 2010*), maintained conserved interactions with the nucleic acid backbone but evolved specificity for A-form duplexes and additional stabilizing interactions with RNA 2′-OH groups in the closed state, as demonstrated here for Mss116. The lack of unwinding activity in some DEAD-box proteins may stem from structural changes in the helicase core that mitigate RNA bending or strand displacement (*Young et al., 2013*). Helicase families that function on DNA (for example, the Swi/Snf, RecQ-like, and UvrD/Rep families) could have diversified by the preservation of conserved interactions with the nucleic acid backbone combined with the selection of additional interactions that favor B-form duplexes and/or disfavor nucleic acids with 2′-OH groups.

Similar inferences can be made from our data about the evolution of distinct mechanisms in SF1 and SF2 families (*Figure 1A*). We propose that although core-closure was retained as a mode of catalysis, the differences in the stability of the closed-state complex between helicase families allowed the diversification of the observed helicase mechanism. Thus, the localized unwinding mechanism used by DEAD-box proteins (*Yang et al., 2007*) likely evolved by the selection of a helicase core that is able to 'clamp' ssRNA and form a highly stable closed-state complex (*Figure 5D,E*). This mode of interaction compensates for the energy cost to locally unwind an RNA duplex, which is critical for DEAD-box protein function (*Del Campo et al., 2009*). In comparison, helicase cores that diverged to form less stable, more transient closed states with ssRNA or ssDNA would favor a mechanism that involved loading and translocating along a single strand (for example, NS3/NPH-II and RecQ-like helicases; *Figure 1A*).

Our data also demonstrate that the stability of the closed state depends upon interactions with nucleotides as well as nucleic acids (*Figure 2*). The DEAH family of helicases are a potential example of a case where a sequence change in motif II compared to DEAD-box proteins ('DEAH' instead of 'DEAD') might result in a weaker interaction with the ATP γ-phosphate and favor the observed switch from localized to translocation-based unwinding (*Figure 1A*). More generally, ATP-dependent core closure to form a ternary complex with nucleic acid may have evolved from tighter to weaker binding as the helicase mechanism concurrently evolved from localized to translocation-based. This is in addition to structural features, such as extra terminal domains or β-hairpins within the helicase core, which favor translocation-based unwinding in some helicase families (*Fairman-Williams et al., 2010*). Protein cofactors may also play a role in helicase substrate specificity, as illustrated for the DEAD-box protein

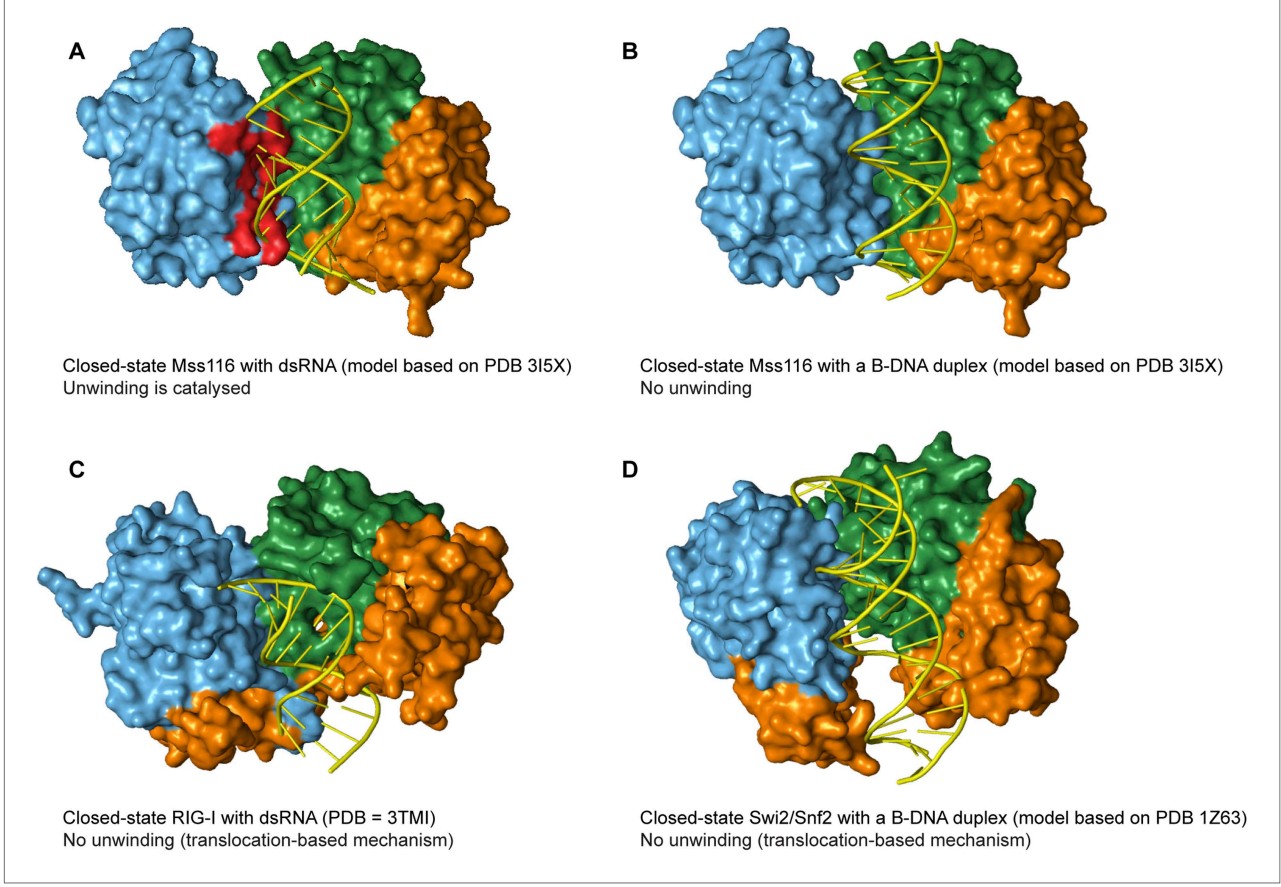

**A**

Closed-state Mss116 with dsRNA (model based on PDB 3I5X)
Unwinding is catalysed

**B**

Closed-state Mss116 with a B-DNA duplex (model based on PDB 3I5X)
No unwinding

**C**

Closed-state RIG-I with dsRNA (PDB = 3TMI)
No unwinding (translocation-based mechanism)

**D**

Closed-state Swi2/Snf2 with a B-DNA duplex (model based on PDB 1Z63)
No unwinding (translocation-based mechanism)

**Figure 7**. Models and crystal structures of closed-state complexes of SF2 helicases. (**A**) Surface representation of closed-state Mss116 with dsRNA modeled in the duplex RNA-binding pocket of D2. Sterically incompatible regions of D1 are highlighted in red, and these indicate how D1 promotes RNA unwinding upon core closure by disrupting the base pairing in the dsRNA. In particular, helicase motifs Ia, Ib, and Ic and the DEAD-box specific post-II motif in D1 displace one RNA strand and bend the other during RNA duplex unwinding (**Mallam et al., 2012**). (**B**) Surface representation of closed-state Mss116 with a B-DNA duplex, which is longer and thinner than an A-form duplex (**Dickerson et al., 1982**), modeled in the duplex RNA-binding pocket of D2. There are no appreciable clashes between dsDNA and the core in this model, which suggests why core closure does not promote unwinding of a B-DNA duplex (**Figure 5C** and **Figure 5—figure supplement 3**). (**C**) Closed-state structure of D1-D3 of human RIG-I helicase (PDB = 3TMI) bound to dsRNA (**Jiang et al., 2011**). dsRNA is accommodated in the closed-state of RIG-I, which explains how it functions by binding and/or translocating along a duplex RNA substrate (**Myong et al., 2009**; **Rawling and Pyle, 2014**). (**D**) Closed-state model of *Sulfolobus solfataricus* Swi2/Snf2 helicase core and a B-DNA duplex adapted from **Durr et al. (2005)**. This model suggests that the Swi2/Snf2 helicase core can accommodate a B-form DNA duplex in a closed-state conformation and explains how helicases in this family function by translocating along DNA duplexes (**Figure 1A**). Proteins and nucleic acids are colored as in **Figure 1**.

Rok1, whose cofactor Rrp5 increases the specificity of the helicase core 10-fold for a pre-rRNA duplex (**Young et al., 2013**).

Finally, other SF2 helicases have evolved to optimally accommodate dsRNA (e.g., RIG-I) or dsDNA (e.g., *Sulfolobus solfataricus* Swi2/Snf2) in a closed state complex and translocate with no observable unwinding (**Figure 1A**, **Figure 7C–D**) (**Durr et al., 2005**; **Myong et al., 2009**; **Jiang et al., 2011**). In these cases, subtle changes in the closed-state core, perhaps combined with additional flanking domains, enable the helicase to bind duplex nucleic acid without the need to overcome the energetic barrier to unwinding and lead to this distinct mechanism of action. It has been hypothesized that during evolution, progenitor enzymes of low activity and broad specificity diverge into families of more potent and highly specialized enzymes (**Jensen, 1976**; **Khersonsky and Tawfik, 2010**). Taken together, our findings suggest how a progenitor helicase core that had broad specificity and used conserved motifs to recognize the phosphate groups of NTPs and the backbone of nucleic acids diverged to present day SF1 and SF2 helicases with different cellular functions.

# Materials and methods

## Oligonucleotides

Unlabeled self-complementary RNA or DNA oligonucleotides (Integrated DNA Technologies, IDT, Coralville, IO; *Figure 4A–C*) were annealed to form 12-bp RNA or DNA duplexes by heating solutions at 6 mM single strands in 100 mM potassium acetate, 30 mM HEPES (pH 7.5) at 94°C for 1 min and then slowly cooling to room temperature over 1 hr. Labeled duplexes for unwinding and binding assays were annealed similarly at 200 µM single strands. Sequences for 12-bp dsDNA substrates were chosen based upon previous studies which indicated that they adopt either A-form or B-form geometry (*Basham et al., 1995*; *Kypr et al., 2009*). We further characterized these substrates by using circular dichroism (CD) to confirm that they retained the required duplex geometry under our experimental conditions in the absence and presence of protein (*Figure 4D,E* and *Figure 4—figure supplement 1*).

## Protein expression and purification

The helicase core of Mss116 (D1D2) and separate domains D1 and D2 were expressed as N-terminal MalE fusions in *Escherichia coli* Rosetta 2 (EMD Biosciences, Germany), grown in ZYP-5052 auto-inducing medium for 24 hr at 22°C, and purified at 4°C, as described (*Del Campo and Lambowitz, 2009*; *Mallam et al., 2011*, *2012*). Proteins for binding and unwinding assays were exchanged into a storage buffer of 20 mM Tris–HCl (pH 7.5), 200 mM KCl, 1 mM dithiothreitol (DTT), 10% glycerol during a final SEC purification step. D1D2 for crystallization was dialyzed into 10 mM Tris–HCl (pH 7.5), 250 mM NaCl, 1 mM DTT, 50 mM arginine + glutamine, 50% glycerol. All proteins were stored at −80°C before use.

## Duplex-unwinding assays in the presence of nucleotide

Equilibrium unwinding of 12-bp dsRNA, A-form DNA, and B-form DNA duplexes was measured in increasing concentrations of NDP-BeF$_x$ (N = A, C, G, or U) using a gel-based fluorescence assay to monitor the formation of a closed-state complex containing a bound single-stranded substrate. Duplexes were labeled with a fluorescent probe (FAM) and quencher (Iowa Black FQ) at the 5′ and 3′ ends, respectively. These substrates gave a change in fluorescence upon unwinding and formation of a closed state (*Figure 2—figure supplement 1A*). NDP-BeF$_x$ (N = A, C, G, or U) was prepared as described (*Del Campo and Lambowitz, 2009*). Measurements were performed using MBP-tagged D1D2 to increase protein solubility under the experimental conditions. MBP-D1D2 (2 µM) was incubated with the appropriate duplex substrate (100 nM) and increasing concentrations of NDP-BeF$_x$-Mg$^{2+}$ (ranging from 0 to 20 mM) at 22°C for at least 1 hr in a reaction medium containing 20 mM Tris–HCl (pH 7.5), 100 mM KCl, 10% glycerol, 1 mM DTT, 5 mM MgCl$_2$, and 0.1 mg/ml of bovine serum albumin. The protein concentration was chosen so that all of the duplex substrate is bound in the open state at equilibrium (*Figure 5A*). Samples were analyzed in a non-denaturing 6% polyacrylamide gel run at 4°C for 60 min. The fluorescence signal of the bound duplex substrate was quantified by using a Typhoon imager (GE Healthcare, UK) to measure the formation of a closed-state complex containing a single-stranded nucleic acid region, indicating duplex unwinding (*Figure 2—figure supplement 1*). The apparent fraction of unwound duplex at increasing concentrations of NDP-BeF$_x$ was quantified by using ImageJ and fit to a one-site binding model to estimate the concentration of nucleotide at the midpoint ($K_{1/2}$) of the unwinding reaction. In all cases, equilibrium was verified by additional assays for samples that were incubated for extended times (up to approximately 4 hr), which gave the same unwinding profiles as those incubated for 1 hr.

Kinetic-unwinding assays of 12-bp dsRNA, A-form DNA, and B-form DNA duplexes by the helicase core were performed with the same fluorophore–quencher labeled probes (*Figure 2—figure supplement 1A*) in the presence of 5 mM NTP (N = A, C, G, or U). In these assays, a change in the fluorescence of the labeled duplex was seen upon unwinding and subsequent re-annealing to form a duplex with an unlabeled strand of the same sequence without a quencher present in excess (*Figure 2—figure supplement 2*). Annealing of these duplexes occurs within the dead time of mixing at the concentration of substrates used in these experiments. D1D2 (2 µM) was mixed with NTP-Mg$^{2+}$ (5 mM), labeled duplex (125 nM), and unlabeled duplex (500 nM) at 22°C in a reaction medium containing 20 mM Tris–HCl (pH 7.5), 100 mM KCl, 10% glycerol, 1 mM DTT, 5 mM MgCl$_2$. Reactions were terminated at appropriate time points with 1 volume of stop buffer (50 mM EDTA, 1% SDS, 10% glycerol) and run in a non-denaturing 20% polyacrylamide at 22°C for 60 min. The fluorescence signal of duplex substrate was quantified by using a Typhoon imager (GE Healthcare) to measure the extent of unwinding/re-annealing.

The apparent fraction of unwound duplex at various time points was quantified by using ImageJ and (where appropriate) fit to a first-order reaction to estimate an observed first-order rate constant ($k_1$).

## Single strand nucleic acid binding assays in the presence of nucleotide

Equilibrium binding of $A_{10}$-RNA and $A_{10}$-DNA to D1D2 in increasing concentrations of NDP-BeF$_x$ was measured by fluorescence anisotropy using MBP-tagged protein to increase the change in anisotropy upon binding. 5′ FAM-labeled $A_{10}$-RNA or $A_{10}$-DNA (10 nM; IDT) was incubated with protein (2 µM) and increasing concentrations of NDP-BeF$_x$ (N = A, C, G, or U; 0 to 10 mM) at 22°C for at least 1 hr in a reaction medium containing 20 mM Tris–HCl (pH 7.5), 100 mM KCl, 10% glycerol, 1 mM DTT, 5 mM MgCl$_2$, and 0.1 mg/ml of bovine serum albumin. The observed fluorescence anisotropy at increasing concentrations of protein was measured by using an EnVision Microplate Reader (Perkin Elmer, Waltham, MA) and was fit to a one-site binding model with a Hill coefficient to estimate the $K_d$ of single-stranded nucleic acid in the presence of increasing nucleotide. Equilibrium was verified by carrying out assays on samples incubated for extended times up to 4 hr, which gave the same binding profiles as those incubated for 1 hr. Equivalent experiments were performed to measure the binding of $A_{10}$-RNA to D1D2 in increasing concentrations of AMP-PNP or ADP (0–10 mM) and ADP + P$_i$ (0–100 mM P$_i$ in the presence of 10 mM ADP).

## Duplex binding assays

Equilibrium binding of 12-bp RNA (A-form) and DNA (A-form and B-form) duplexes to D1 or D2 was measured by EMSA using MBP-tagged proteins to increase protein solubility as described (*Mallam et al., 2012*). 5′ FAM-labeled 12-bp duplexes (100 nM; IDT; *Figure 4A–C*) were incubated with increasing concentrations of protein (0–6 µM) at 22°C for at least 1 hr in a reaction medium containing 20 mM Tris–HCl (pH 7.5), 100 mM KCl, 10% glycerol, 1 mM DTT, 5 mM MgCl$_2$, and 0.1 mg/ml of bovine serum albumin to stabilize the protein at low concentrations. Samples were then analyzed in a non-denaturing 6% polyacrylamide gel run at 4°C for 60 min, and the fluorescence signal of the bound duplex substrate was quantified by using a Typhoon imager. The fraction of bound duplex with increasing concentrations of MBP-tagged protein was quantified by using ImageJ and fit to a one-site binding model with a Hill coefficient to estimate a $K_d$.

Competition assays were performed similarly by measuring the competitive displacement from MBP-D2 (500 nM) of 5′ FAM-B-DNA duplex (250 nM) by unlabeled dsRNA (0–6 µM, $K_i$ = 860 ± 40 nM) and of 5′ FAM-dsRNA (250 nM) by unlabeled B-DNA duplex (0–6 µM, $K_i$ = 1700 ± 200 nM). In these cases, the fraction of free substrate was quantified and a $K_i$ was estimated from a one-site binding model.

## Size-exclusion chromatography

Binding of nucleotide and nucleic acid substrates to D1D2 was examined by size-exclusion chromatography. The helicase core of Mss116 does not contain tryptophan residues and its calculated extinction coefficient is small ($\varepsilon_{280}$ = 18,255 M$^{-1}$ cm$^{-1}$; ExPASy Proteomics Server ProtParam tool [*Wilkins et al., 1999*]). The formation of a closed-state complex in the presence of nucleic acid and NDP-BeF$_x$ therefore gives rise to a large change in $A_{260}$ compared to $A_{280}$. Protein samples (10 µM) were incubated at 22°C for 30 min in NDP-BeF$_x$-Mg$^{2+}$ (5 mM, N = A, C, G, or U) and single-stranded ($A_{10}$-RNA or $A_{10}$-DNA; 20 µM) or duplex (dsRNA, A-DNA duplex or B-DNA duplex; 10 µM) nucleic acid and loaded onto a Superdex 75 column (GE Healthcare) pre-equilibrated in a buffer containing 20 mM Tris–HCl (pH 7.5), 200 mM KCl, 10% glycerol, 1 mM DTT, 5 mM MgCl$_2$. The absorbance and elution volume of the protein complexes above the background signal of the buffer were measured at 260 and 280 nm (*Table 1*). Control samples of protein alone, substrate alone, or protein and either nucleotide or nucleic acid were also measured; closed-state complexes were not detected in these cases.

## Circular dichroism

All measurements were performed in 20 mM Tris–HCl (pH 7.5), 100 mM KCl, 10% glycerol, 1 mM DTT, 5 mM MgCl$_2$ buffer using a thermostatically controlled 0.01-cm path-length cuvette at 25°C and a Jasco J-815 spectrometer (Jasco Inc., Easton, MD). Scans were taken between 200 and 325 nm at a scan rate of 0.5 nm s$^{-1}$ with 30 accumulations. Measurements were made on samples of SEC-purified A-form DNA or B-form DNA duplexes (100 µM) in the absence or presence of Mss116 D2 or MBP-D2 (120 µM).

## Crystallization

For the D1D2–$A_{10}$-RNA–NDP-BeF$_x$ complexes, protein (~350 μM) was incubated with $A_{10}$-RNA (600 μM), NDP-BeF$_x$-Mg$^{2+}$ (5 mM; N = A, C, G, or U) and MgCl$_2$ (1 mM) for 30 min on the desktop. Sitting drops were assembled using 0.5 μl of complex and 0.5 μl of a well solution of 0.1 M HEPES, pH 7.5, 2% tacsimate, pH 7.0, 20% PEG (polyethylene glycol) 3350 for D1D2–$A_{10}$-RNA–ADP-BeF$_x$; 0.2 M sodium malonate, pH 5.0, 20% PEG 3350 for D1D2–$A_{10}$-RNA–CDP-BeF$_x$; 4% tacsimate, pH 8.0, 12% PEG 3350 for D1D2–$A_{10}$-RNA–GDP-BeF$_x$; and 0.1 M DL-malic acid, pH 7.0, 12% PEG 3350 for D1D2–$A_{10}$-RNA–UDP-BeF$_x$ (Hampton Research, Aliso Viejo, CA). Drops were stored at 22°C and plate-like crystals appeared within 1–2 weeks. Crystals were removed from sitting drops and flash cooled immediately in liquid $N_2$. Crystals of D1D2–$A_{10}$-DNA–ADP-BeF$_x$ were obtained similarly and drops were assembled with a well solution of 0.2 M ammonium acetate, 20% PEG 3350.

## Structure determination

X-ray diffraction data were collected at the Advanced Light Source (ALS), Lawrence Berkeley National Laboratory (mail-in service on beamlines 5.0.2 or 5.0.3; wavelength = 1.00003 Å). Details of data collection and refinement are in *Table 2*. Diffraction intensities were indexed, integrated, and scaled with HKL-2000 (*Otwinowski and Minor, 1997*). Initial space groups were determined by using Pointless (*Evans, 2006*) and confirmed by decreases in both $R_{work}$ and $R_{free}$ after refinement of molecular replacement solutions. Molecular replacement was performed with Phaser (*McCoy et al., 2007*), using the previously determined structure of Mss116 D1D2 in the closed state (PDB 3I5X) as a search model. Structures were completed with cycles of manual model building in Coot (*Emsley et al., 2010*) and refinement in Phenix (*Adams et al., 2010*). Validation of protein and nucleic acid models and their contacts was done by using MolProbity (*Chen et al., 2010*) and indicated that at least 98% of residues are located in the most favorable region of the Ramachandran plot. Structural figures were prepared by using the PyMOL Molecular Graphics System, Version 1.4, Schrödinger, LLC.

## Accession numbers

Coordinates and structure factors were deposited in the Protein Data Bank under accessions 4TYW (D1D1–$A_{10}$-RNA–ADP-BeF$_x$), 4TYY (D1D1–$A_{10}$-RNA–CDP-BeF$_x$), 4TZ0 (D1D1–$A_{10}$-RNA–GDP-BeF$_x$), 4TZ6 (D1D1–$A_{10}$-RNA–UDP-BeF$_x$), and 4TYN (D1D1–$A_{10}$-DNA–ADP-BeF$_x$).

## Acknowledgements

We thank A Monzingo (University of Texas at Austin Macromolecular Crystallography Facility) for help with X-ray diffraction data collection and R Russell (University of Texas at Austin) for comments on the manuscript. X-ray diffraction data were collected at the Berkeley Center for Structural Biology, which is supported in part by the NIH, NIGMS, and HHMI. The Advanced Light Source is supported by the Director, Office of Science, Office of Basic Energy Sciences, of the U.S. Department of Energy under Contract No. DE-AC02-05CH11231. This work was supported by NIH grant GM037951.

## Additional information

### Funding

| Funder | Grant reference number | Author |
|---|---|---|
| National Institutes of Health | GM037951 | Alan M Lambowitz |

The funder had no role in study design, data collection and interpretation, or the decision to submit the work for publication.

### Author contributions

ALM, Conception and design, Acquisition of data, Analysis and interpretation of data, Drafting or revising the article; DJS, Analysis and interpretation of data, Drafting or revising the article; AML, Conception and design, Analysis and interpretation of data, Drafting or revising the article

## Additional files

### Major datasets

The following datasets were generated:

| Author(s) | Year | Dataset title | Dataset ID and/or URL | Database, license, and accessibility information |
|---|---|---|---|---|
| Mallam AL, Sidote DJ, Lambowitz AM | 2014 | DEAD-box helicase Mss116 bound to ssRNA and ADP-BeF | http://www.pdb.org/pdb/search/structidSearch.do?structureId=4TYW | Publicly available at RCSB Protein Data Bank. |
| Mallam AL, Sidote DJ, Lambowitz AM | 2014 | DEAD-box helicase Mss116 bound to ssRNA and CDP-BeF | http://www.pdb.org/pdb/search/structidSearch.do?structureId=4TYY | Publicly available at RCSB Protein Data Bank. |
| Mallam AL, Sidote DJ, Lambowitz AM | 2014 | DEAD-box helicase Mss116 bound to ssRNA and GDP-BeF | http://www.pdb.org/pdb/search/structidSearch.do?structureId=4TZ0 | Publicly available at RCSB Protein Data Bank. |
| Mallam AL, Sidote DJ, Lambowitz AM | 2014 | DEAD-box helicase Mss116 bound to ssRNA and UDP-BeF | http://www.pdb.org/pdb/search/structidSearch.do?structureId=4TZ6 | Publicly available at RCSB Protein Data Bank. |
| Mallam AL, Sidote DJ, Lambowitz AM | 2014 | DEAD-box helicase Mss116 bound to ssDNA and ADP-BeF | http://www.pdb.org/pdb/search/structidSearch.do?structureId=4TYN | Publicly available at RCSB Protein Data Bank. |

The following previously published datasets were used:

| Author(s) | Year | Dataset title | Dataset ID and/or URL | Database, license, and accessibility information |
|---|---|---|---|---|
| Del Campo M, Lambowitz AM | 2009 | Structure of Mss116p bound to ssRNA and AMP-PNP | http://www.pdb.org/pdb/explore/explore.do?structureId=3I5X | Publicly available at RCSB Protein Data Bank. |
| Jiang F, Ramanathan A, Miller MT, Tang GQ, Gale M, Patel SS, Marcotrigiano J | 2011 | Structural Basis for RNA Recognition and Activation of RIG-I | http://www.pdb.org/pdb/explore/explore.do?structureId=3TMI | Publicly available at RCSB Protein Data Bank. |

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
