## [Decision Letter]

Thank you for sending your work entitled “Molecular insights into RNA and DNA
helicase evolution from the determinants of specificity for a DEAD-box RNA
helicase” for consideration at *eLife*. Your article has been
favorably evaluated by James Manley (Senior editor), a Reviewing editor and 2
reviewers.

Leemor Joshua-Tor (Reviewing editor) and Ben Luisi (peer reviewer) have agreed to reveal
their identity.

The Reviewing editor and the reviewers discussed their comments before we reached this
decision, and the Reviewing editor has assembled the following comments to help you
prepare a revised submission.

Lambowitz and co-workers address an important question in RNA biology: how RNA helicases
have become specific for RNA, even though they are structurally very similar to DNA
helicases. Using structural and biochemical approaches on the DEAD-box helicase Mss116
that functions as a chaperone for the process of mitochondrial intron splicing, they
show that this helicase can, in principle, bind both RNA and DNA, and a variety of
nucleotides, but with a significant preference for A over the others. Crystal structures
are presented of the Mss116 construct in complex with Berylium Fluoride and ADP, CDP,
GDP and UDP. These structures go some way to delineate specific interactions that confer
specificity for the adenine base and RNA, and show that these specificities are caused
by rather small tweaks in the helicase core. Their results also suggest that the gamma
phosphate of ATP pays a critical role in maintaining a 'closed state'
conformation when engaging the nucleotide and RNA. The authors also show that, in
addition to the known ability to bind duplex RNA, Mss116 is capable of binding to A-form
double stranded DNA, but not DNA in the more common B-form, while the isolated
C-terminal RecA domain (D2) can bind the B-form. Crystal structures are also presented
of Mss116 in complex with A10 RNA and A10 DNA, showing a similar mechanism of
interaction for both substrates.

In general, the paper is well written and the methodology is sound, and the experimental
work, encompassing X-ray crystallography and in vitro binding and unwinding assays, is
comprehensive. The Discussion is instructive and insightful. The figures are excellent
and are very easy on the eye. Although the results are perhaps not ground breaking, the
paper provides new, and in some aspects surprising, insight for a large and ubiquitous
family of enzymes with central biological roles. The manuscript will thus be of interest
to a diverse readership. There are a number of minor comments on specific points that
will hopefully be helpful for the authors to consider, outlined below.

1) Is it clear that Mss116 functions in vivo in isolation, or is the enzyme's
activity directed and dependent upon other partner proteins? How might this bear on the
central conclusion of the Abstract and Discussion, regarding the apparent
family-specific specificity? Many of the DExD-box proteins seem to have partners that
strongly affect their activity.

2) A surprising finding is the ability of Mss116 to unwind pure DNA duplexes, provided
these adopt A-form geometry. The unwinding is shown with ADP-BeFx, a non-hydrolysable
ATP analog. It would be important to check whether unwinding is also seen with ATP, as
several recent papers suggest that DEAD-box helicases, which have previously been
considered as pure RNA helicases, might function on DNA.

3) Unwinding of RNA with all four nucleotide diphosphate beryllium fluoride compounds is
also surprising. Here, again, it would be instructive to test unwinding with the
nucleotide triphosphates. These results might be influenced by the affinity of binding
to the nucleotide rather than the ability of the helicase to hydrolyse a particular
nucleotide and simultaneously unwind the duplex, for example. In addition, the different
triphosphates have been tested in unwinding assays for many DEAD-box helicases, and it
appears that no unwinding has been reported with NTPs other than ATP. It would thus be
important to sort out whether one should have tested these NTPs over a larger
concentration range than examined, or whether the use of the diphosphate beryllium
fluoride compounds somehow biases these experiments, as has been suggested, too. If a
bias were to be found, this would not invalidate any of the points made, but add a
further important facet to the story.

4) The authors report K_d_ values for ADP-BeFx binding to Mss116 in the
micromolar range. Binding of ADP-BeFx to Mss116 and RNA has recently been described by
Liu et al. (Biochemistry, 2014, 53, 423–433). Although no K_d_ values
were reported there, the ADP-BeFx complex with Mss116 was very long-lived. It might be
prudent to specifically verify that equilibrium was reached for the K_d_
measurements, even after incubation of 1h. In addition, the paper by Liu et al. should
probably be cited.

5) In Figure 1 it might be helpful to extend the
cartoon to include the final step in the unwinding cycle where the ATP is hydrolysed,
and then the helicase re-enters the open state upon departure of the ADP, Pi and
RNA.

6) In Figure 2, perhaps it would be better to
have the binding assay (Figure B) first, then the unwinding assay (A).

7) Figure 3: Label G128 in Figure B? Is the Q133
oriented such that the carbonyl is forced to be pointed at the O6 of the GDP? This would
be an unfavorable interaction.

8) Results section, “The structural basis for the ATP specificity of the helicase
core of Mss116”: “two side-chain hydrogen (H)-bonds from G128 and
E133”. Figure 3 has residue 133 labelled
as Q. Which one is it? Also, can glycine really make a side chain H-bond?

9) Summary: 'core stability'. Perhaps change the wording here, because as
written this might be misinterpreted as stability of the fold, which is clearly not
intended.

10) Results section (“analytical size-exclusion chromatography (SEC) shows that a
94 closed-state helicase…”), Table 1 and Figure 5—figure supplement 2: do the elution volumes definitively demonstrate whether the state is open
or closed? There is for example the formal possibility that the proteins are forming
oligomers. Do the authors have access to SEC-MALS? This would give the molecular weights
fairly precisely, which can be used to confirm that the complexes are intact.

11) Results section: the sentence “the effective concentration of the ATP gamma
phosphate plays a key role in maintaining the closed-state structure” appears to
be repetitive with “the effective concentration of the ATP gamma phosphate is
critical for the stability of the closed-state”.

12) Table 2: the angles for the unit cell are
not in Greek characters.

13) Figure 2—figure supplement 1 appears
to be missing.

14) Figure 4: out of curiosity, what happens to
the binding of the A-form DNA with D2, does it retain A-form CD spectral signature?

---

## [Author Response]

*1) Is it clear that Mss116 functions in vivo in isolation, or is the
enzyme's activity directed and dependent upon other partner proteins? How might
this bear on the central conclusion of the Abstract and Discussion, regarding the
apparent family-specific specificity? Many of the DExD-box proteins seem to have
partners that strongly affect their activity*.

Mss116 by itself displays high unwinding activity in the absence of partner proteins in
vitro, and functional studies give no indication that Mss116’s helicase activity
is directed by or dependent upon a partner protein in vivo. Instead, Mss116 functions as
a general RNA chaperone that binds diverse RNA and RNP substrates non-specifically and
resolves kinetic traps that impede RNA folding (15; 6).
In vivo, Mss116 is required for the efficient splicing of all 13 yeast mitochondrial
group I and group II introns, as well as the translation of certain mRNAs and
mitochondrial RNA processing reactions (15). The introns whose splicing is promoted by Mss116 differ structurally
and rely on different intron-encoded maturases or nuclear gene-encoded splicing factors
for structural stabilization. In vitro, purified Mss116 can by itself promote splicing
of group II introns or function in the presence of a heterologous splicing factor, the
*Neurospora crassa* mitochondrial tyrosyl-tRNA synthetase (CYT-18
protein), to promote the splicing of a *N. crassa* group I intron (6). The lack of specificity
required for Mss116 function in vivo is further indicated by the finding that all
phenotypic defects in an *mss116*Δ strains can be rescued by the
expression of the *N. crassa* CYT-19 protein (15), a related DEAD-box that also functions as
general RNA chaperone (Mohr et al., 2002).

In addition to its function as a general RNA chaperone, Mss116 co-purifies with the
yeast mt RNA polymerase (Markov et al., 2009) and may function as a transcription
elongation factor for this enzyme (Markov et al., 2014). However, this function does not
require the helicase activity (i.e., the ATPase or RNA-unwinding activities) of
Mss116.

To address this comment, we have modified the Introduction to state that Mss116 binds
diverse RNA substrates non-specifically and displays high RNA helicase activity in the
absence of partner proteins.

*2) A surprising finding is the ability of Mss116 to unwind pure DNA duplexes,
provided these adopt A-form geometry. The unwinding is shown with ADP-BeFx, a
non-hydrolysable ATP analog. It would be important to check whether unwinding is also
seen with ATP, as several recent papers suggest that DEAD-box helicases, which have
previously been considered as pure RNA helicases, might function on DNA*.

We performed additional kinetic unwinding assays, which demonstrate that Mss116 can use
ATP to unwind the RNA duplex and the A-DNA duplex substrates with observed first-order
rate constants (*k*_1_) of 0.46 and 0.15 min^-1^,
respectively. However, we do not observe unwinding of the B-DNA duplex with ATP. This is
the same unwinding trend that we observed with ADP-BeF_x_. The kinetic
unwinding assays have been added to the Results (Figure 5—figure supplement 3) and to the Methods sections, along with
references to recent papers suggesting DEAD-box helicases might function on DNA (21; 43).

*3) Unwinding of RNA with all four nucleotide diphosphate beryllium fluoride
compounds is also surprising. Here, again, it would be instructive to test unwinding
with the nucleotide triphosphates. These results might be influenced by the affinity
of binding to the nucleotide, rather than the ability of the helicase to hydrolyse a
particular nucleotide and simultaneously unwind the duplex, for example. In addition,
the different triphosphates have been tested in unwinding assays for many DEAD-box
helicases, and it appears that no unwinding has been reported with NTPs other than
ATP. It would thus be important to sort out whether one should have tested these NTPs
over a larger concentration range than examined, or whether the use of the
diphosphate beryllium fluoride compounds somehow biases these experiments, as has
been suggested, too. If a bias were to be found, this would not invalidate any of the
points made, but add a further important facet to the story*.

We performed kinetic unwinding assays, as in point 2, to compare the unwinding of the
12-bp RNA duplex catalyzed by ATP to CTP, GTP and UTP, which are documented in the
Methods and reported in the Results (Figure 2—figure supplement 2). These assays demonstrate that NTPs other than
ATP do not promote unwinding of the RNA duplex even with the other NTPs added at 5 mM
concentration. We also performed these kinetic unwinding assays in a different buffer
with 0.5 mM free Mg^2+^, as previous data indicates that the unwinding
activity of Mss116 increases at lower Mg^2+^ concentrations (13). However, we still found no
unwinding for CTP, GTP or UTP under these conditions. These findings suggest that the
closed state with ssRNA and nucleotide triphosphates other than ATP does not form and
catalyse the unwinding of the RNA duplex used in these assays. Use of the diphosphate
beryllium fluoride analogues is therefore necessary to access and stabilize the
closed-states that promote unwinding with other nucleotide bases.

*4) The authors report K*_*d*_
*values for ADP-BeFx binding to Mss116 in the micromolar range. Binding of
ADP-BeF_x_ to Mss116 and RNA has recently been described by Liu et al.
(Biochemistry, 2014, 53, 423–433). Although no
K*_*d*_
*values were reported there, the ADP-BeF_x_ complex with Mss116 was very
long-lived. It might be prudent to specifically verify that equilibrium was reached
for the K*_*d*_
*measurements, even after incubation of 1h. In addition, the paper by Liu et al.
should probably be cited*.

We had verified that equilibrium was reached in all our experiments by carrying out
assays for extend times (up to approximately 4 h), which gave the same binding or
unwinding profiles as those incubated for 1 h. This information has been incorporated
into the Methods section. We note that the protein (full-length Mss116 in Liu et al.
versus the helicase core in our study), RNA substrates, and reaction conditions differ
between the two studies. We have cited the paper by Liu et al. as example of the
different behaviors of ATP analogues.

*5) In*
Figure 1
*it might be helpful to extend the cartoon to include the final step in the
unwinding cycle where the ATP is hydrolysed, and then the helicase re-enters the open
state upon departure of the ADP, P_i_ and RNA*.

We have added this step final step in the unwinding cycle to Figure 1.

*6) In*
Figure 2*, perhaps it
would be better to have the binding assay (Figure B) first, then the unwinding assay
(A)*.

In the current version of the manuscript, the panels that display the unwinding (Figure 2) and binding (Figure 2) appear in the figure in the order that they are
mentioned in the text.

*7)*
Figure 3*: Label G128 in
Figure B? Is the Q133 oriented such that the carbonyl is forced to be pointed at the
O6 of the GDP? This would be an unfavorable interaction*.

The O-O distance from the carbonyl of the Q133 side chain and the O6 of the GDP is 3.4
Å in the GDP structure. This is compared to 2.4 Å in the ADP structure, even
though the two distances look similar in the view in Figure 3. The carbonyl of Q133 is therefore not pointing directly towards
the O6 of the GDP, and the only favorable contact with this base is made by Q133 to
N7.

*8) Results section, “The structural basis for the ATP specificity of the
helicase core of Mss116”: “two side-chain hydrogen (H)-bonds from G128
and E133”.*
Figure 3
*has residue 133 labelled as Q. Which one is it? Also, can glycine really make a
side chain H-bond*?

We have corrected this sentence to read: ‘two hydrogen (H)-bonds from G128 and
Q133…”.

*9) Summary: 'core stability'. Perhaps change the wording here, because
as written this might be misinterpreted as stability of the fold, which is clearly
not intended*.

We have changed ‘core stability’ to ‘complex stability’.

*10) Results section (“analytical size-exclusion chromatography (SEC)
shows that a 94 closed-state helicase…”), Table1 and*
Figure 5—figure supplement 2*: do the elution volumes definitively demonstrate whether
the state is open or closed? There is for example the formal possibility that the
proteins are forming oligomers. Do the authors have access to SEC-MALS? This would
give the molecular weights fairly precisely, which can be used to confirm that the
complexes are intact*.

The ratio of A_260_/A_280_, which is approximately 0.6 for free
protein and >1 for protein-nucleic acid complexes, was used as an indicator of the
formation of a closed-state complex that contains nucleotide and nucleic acid. Protein
cores that remain mostly in the open state under the SEC conditions have a
A_260_/A_280_ ratio much closer to that of the free protein. The
elution volumes are consistent with the complex formation and are used only in
conjunction with the A_260_/A_280_ ratio as indicating the open or
closed state of the core.

*11) Results section: the sentence “the effective concentration of the ATP
gamma phosphate plays a key role in maintaining the closed-state structure”
appears to be repetitive with “the effective concentration of the ATP gamma
phosphate is critical for the stability of the closed-state”*.

We have removed the sentence: “They also indicate that the effective
concentration of the ATP gamma phosphate plays a key role in maintaining the
closed-state structure”.

*12)*
Table 2*: the angles
for the unit cell are not in Greek characters*.

We have changed the angles of the unit cell to Greek characters.

*13)*
Figure 2—figure supplement 1
*appears to be missing*.

Apologies for the omission of this Figure supplement, it is now included.

*14)*
Figure 4*: out of
curiosity, what happens to the binding of the A-form DNA with D2, does it retain
A-form CD spectral signature*?

We have now also measured the CD spectra of the A-form DNA duplex when bound to Mss116
D2, and this duplex also retains A-form geometry upon binding. This is included as Figure 4—figure supplement 1.